# PowderMEMS—A Generic Microfabrication Technology for Integrated Three-Dimensional Functional Microstructures

**DOI:** 10.3390/mi13030398

**Published:** 2022-02-28

**Authors:** Thomas Lisec, Ole Behrmann, Björn Gojdka

**Affiliations:** Fraunhofer Institute for Silicon Technology ISIT, Fraunhoferstr. 1, 25524 Itzehoe, Germany; thomas.lisec@isit.fraunhofer.de (T.L.); ole.behrmann@isit.fraunhofer.de (O.B.)

**Keywords:** MEMS integration, three-dimensional microstructures, powder-based microstructures, porous MEMS, micromagnets, microfluidics, energy harvesting, flow sensors, gas sensors, integrated inductors

## Abstract

A comprehensive overview of PowderMEMS—a novel back-end-of-line-compatible microfabrication technology—is presented in this paper. The PowderMEMS process solidifies micron-sized particles via atomic layer deposition (ALD) to create three-dimensional microstructures on planar substrates from a wide variety of materials. The process offers numerous degrees of freedom for the design of functional MEMSs, such as a wide choice of different material properties and the precise definition of 3D volumes at the substrate level, with a defined degree of porosity. This work details the characteristics of PowderMEMS materials as well as the maturity of the fabrication technology, while highlighting prospects for future microdevices. Applications of PowderMEMS in the fields of magnetic, thermal, optical, fluidic, and electrochemical MEMSs are described, and future developments and challenges of the technology are discussed.

## 1. Introduction

The fabrication of 3D microstructures with dimensions between several tens and several hundreds of microns, on planar substrates such as silicon or glass wafers, is of considerable interest for MEMSs. Although subject to continuous research, the established methods are confined in one way or another by the applicable materials, the geometric precision, and various integration issues, such as compatibility with the substrate or the requirements of the MEMS manufacturing process itself.

For subtractive patterning of substrates, wet or dry etching techniques are widely used. The individual methods are often dedicated to specific materials, such as deep reactive-ion etching (DRIE) for silicon [1] or laser-induced deep etching (LIDE) for glass [2]. Laser machining is more universal, and can be applied to polymeric, metallic, or ceramic substrates, but is time-consuming due to its serial nature [3]. In addition, various additive methods are available to add functional 3D microstructures with certain material properties at specific locations on the substrate. Commonly, a film of the required material is deposited onto the substrate and subsequently patterned. While the desired lateral dimensions are easily achievable using this technique, patterning of layers that exceed a few µm in thickness becomes increasingly challenging. For example, materials such as piezoelectric PZT or hard magnetic NdFeB can be obtained with a thickness of 100 µm via aerosol deposition and pulsed-laser deposition, respectively [4,5]. However, suitable patterning processes have been reported only for 10 µm thick layers of those materials [6,7]. With photolithography, patterning of thick layers in the range of several hundred µm can be achieved; however, the material choice is limited to (photosensitive) polymers [8]. Other techniques with intrinsic patterning—such as multilayer screen or inkjet printing [9], or electroplating [10,11]—have similar restrictions.

Limitations in material choice as well as patterning issues can be avoided using powder-based techniques for 3D microstructure fabrication. Widely used is the filling of mold patterns with particle-loaded inks or pastes via spin coating or squeegee coating. The properties of such 3D microstructures can be tailored within a broad range. For example, using Ag powder, conductive lines can be fabricated [12]. Utilizing NdFeB powder, micromagnets can be obtained [13]. Soft magnetic powder has been applied to fabricate integrated inductor cores [14,15]. However, the geometric precision and minimal structural dimensions of these methods are limited. Since the viscosity of inks and pastes rises with the particle load, proper filling of molds with a width of several tens of microns is challenging. In addition, the presence of an organic matrix material within the 3D microstructures limits their thermal stability and corrosion resistance, restricting further processing of such substrates in common MEMS fabrication environments. Powder-based techniques such as selective laser beam sintering [16] or melting [17] allow the generation of organic-free microstructures that do not suffer from low thermal stability. However, although both techniques are widely applied to create free-standing parts, their implementation on planar substrates is a challenge. For example, since sintering and melting processes are always associated with high temperatures and shrinking effects, large mechanical stresses arise at the interface between the substrate and the 3D microstructures [18].

Recently, PowderMEMS—a novel powder-based microfabrication technique—has been proposed [19]. Instead of sintering, welding, or the application of binders, micron-sized particles are solidified into rigid, porous 3D microstructures by means of atomic layer deposition (ALD). Compared to other state-of-the-art processes, this technique fulfills a unique set of requirements for the integration of 3D functional microstructures at the substrate level: a multitude of dielectric, metallic, or semi-conducting materials can be used; structures with thicknesses of several hundreds of micrometers can be obtained; porous and magnetic volumes can be fabricated; and integration is performed at the substrate level via a batch-enabled, low-temperature process. In addition, it has been shown that silicon substrates incorporating such 3D microstructures are compatible with common back-end-of-line (BEOL) and MEMS production environments. Thus, post-processing of substrates containing PowderMEMS microstructures is possible with standard processes of MEMSs and semiconductor technology [19,20]. These unique capabilities of the novel technique enable the improvement of existing MEMSs and the creation of novel devices in all application areas.

This article provides an overview of the state-of-the-art of PowderMEMS, summarizes the morphology of the resulting microstructures, and discusses applications that are currently under development. In Section 2, the generic PowderMEMS process and its implementation are presented. The morphology of the resulting structures is described in Section 3. The innovation potential of PowderMEMS structures is discussed, for various applications, in Section 4.

## 2. The PowderMEMS Process

### 2.1. General Description of the Process and Its Features

The PowderMEMS process comprises three key steps: dry filling of molds with microfine powder, subsequent solidification of the powder via ALD, and conditioning of the substrate for further processing [19]. Figure 1 schematically depicts these steps. Starting materials are planar substrates—e.g., silicon or glass wafers—with pre-structured micro-molds and a dry powder of micron-sized particles of the filling material of choice.

In the first step, the molds are filled with the dry particles, as illustrated in Figure 1a. The utilization of dry particles in combination with a dedicated filling technique [21] ensures a dense and reproducible filling of molds with lateral dimensions from 20 µm up to 4000 µm, and a depth of up to 1000 µm. However, due to its particulate nature, it is not possible to produce structures with a fill factor of 100% using PowderMEMS. Depending on the application, this might either be an advantage (e.g., microfluidics) or a drawback when compared to bulk material (e.g., mechanical stability).

In the second step, the loose particles are subjected to a low-temperature ALD process, as depicted in Figure 1b. Since the gaseous ALD precursors penetrate the voids between neighboring particles, the growing ALD layer homogeneously coats all particles in the particle bed, as well as the inner and outer surfaces of the mold. Neighboring particles are thus connected to one another at the points of contact, forming a mechanically stable porous 3D microstructure throughout the entire mold volume. Note that at this point in the process the structure remains embedded within the substrate, but can be released from the mold using an appropriate etching method (Figure 1b).

In the third and final step, the substrate with the integrated PowderMEMS structures is conditioned to enable further post-processing in a (MEMS-) cleanroom environment. The conditioning comprises the removal of excess particles from the surface of the substrate, as depicted in Figure 1c.

Since ALD is performed at comparatively low temperatures, the PowderMEMS procedure can be applied to a broad variety of powders. Solidification with Al_2_O_3_ is, for example, possible at temperatures between 75 °C and 300 °C via thermal ALD, using TMA and H_2_O as precursors [22]. Apart from Al_2_O_3_, solidification has already been demonstrated with SiO_2_ at 300–350 °C using SIBDEA and ozone [23]. High-aspect-ratio thermal ALD is available for other metal oxides—such as TiO_2_, V_2_O_5_, and ZnO—nitrides, and metals [24], yielding numerous ALD/powder material combinations for functional PowderMEMS structures.

Another advantage is that substrates incorporating PowderMEMS microstructures are compatible with common BEOL and MEMS production environments, and can be post-processed using standard processes of MEMS and semiconductor technology [19,25]. As solidification via ALD is performed at comparatively low temperatures, the stresses between the substrate and the 3D microstructures remain low compared to high-temperature processes such as sintering. Since no pressure is applied to the particle bed during solidification, the mold dimensions are reproduced with high precision, without shrinkage of the PowderMEMS structure. The ALD layer envelops each particle, thus protecting them efficiently from the surrounding process or application environment. Since no organic materials are involved in the PowderMEMS procedure, the 3D microstructures exhibit excellent thermal stability, surviving temperatures up to 425 °C without degradation [20]. To illustrate the integrability of the PowderMEMS procedure in MEMS processes, Figure 2 displays a piezoelectric vibrational energy harvester featuring an array of permanent micromagnets fabricated from NdFeB powder at the wafer level (see Section 4 and [26] for details).

### 2.2. Implementation of the PowderMEMS Process at Fraunhofer ISIT

Successful utilization of PowderMEMS structures in industrially relevant microsystems requires a sufficiently high level of technological maturity. For the PowderMEMS process, this especially demands the following:That all processes are automated, and allow for a sufficient throughput for mass production;That the processes are reproducible, with low variation across individual substrates, as well as from substrate to substrate;That non-destructive and fast characterization and process control methods are available at the substrate level;That the substrates can be appropriately conditioned to allow for post-processing in a common BEOL-compatible cleanroom environment after embedding the 3D microstructures.

Figure 3 illustrates how PowderMEMS microstructures are currently integrated on 8-inch silicon substrates at Fraunhofer ISIT. After creating the micro-mold pattern via DRIE in the MEMS cleanroom, the substrates are transferred into a dedicated laboratory. Here, the dry-filling procedure, the solidification via ALD, and part of the substrate conditioning are performed. The photoresist mask used for mold etching remains on the wafer as a sacrificial layer.

For filling of the micro-molds with particles, a novel automated procedure has been developed [21]. The powder is dry-filled into the cavities using a combination of low-frequency and ultrasound vibration, in conjunction with mechanical compacting of the powder. With this procedure, molds with dimensions down to 10 micrometers can be filled reproducibly. Excess powder is removed from the wafer surface with an automated squeegee.

Subsequently, the substrates are transferred into the ALD tool (Picosun R3000B), in which the loose particles are agglomerated into rigid microstructures. The wafers are placed horizontally within the reaction chamber to prevent the loose powder from falling out of the molds before solidification. Currently, six 8-inch wafers can be processed in a single batch. Deposition of 75 nm Al_2_O_3_ at 75 °C has proven to be a suitable solidification layer for various applications [19].

As the corresponding cross-sections in Figure 3 illustrate, some excess particles remain on the substrate surface on the front side of the substrate, as well as on its backside, after mold filling. During the subsequent agglomeration process, these particles are fixed firmly to the substrate surface by the ALD solidification layer. Therefore, substrate conditioning is carried out after the ALD process. Bottom-side cleaning is performed by grinding and polishing with an automatic grinder (Disco DFG8540), as previously described in [25]. Approximately 25 µm of silicon is typically removed from the bottom side of the substrates. For front-side conditioning, the photoresist mask left after DRIE is used as a sacrificial layer. After transfer of the substrates into the cleanroom, the photoresist is removed by O_2_ plasma etching (Tepla barrel asher), followed by lift-off in an organic solvent and a final high-velocity spray clean (SSEC 3000 automatic spin etcher) [25].

After surface conditioning, the substrates can be further processed in the cleanroom if required. As already mentioned, wafers with embedded PowderMEMS microstructures are BEOL-compatible, i.e., wafer processing can be implemented on common cleanroom equipment without major process changes at process temperatures up to 425 °C. For example, to finalize the piezoelectric vibrational energy harvester shown in Figure 2 after micromagnet integration, post-processing includes various thin-film depositions as well as wet and dry etching of metals, dielectrics, and silicon [10].

For magnetization of integrated magnetic structures, a custom-built wafer-level magnetization tool (MAGSYS, Dortmund, Germany) is available, which provides magnetic fields of up to 3.5 T across a whole 8-inch wafer. After loading the wafer to the tool, the substrate is automatically positioned within a coil. Subsequently, a current of up to 5.6 kA is passed through the coil for generation of the magnetic field.

### 2.3. Process Control

Appropriate methods to control process quality and to support process development are central to the successful use of PowderMEMS for innovative microsystems in both research and production. NdFeB-based micromagnets have proven to be suitable as test structures for process characterization. Vibrating-sample magnetometer (VSM) measurements can be utilized to monitor the reproducibility of the PowderMEMS processes—i.e., micro-mold filling and agglomeration [21]—as well as for overall process development [20,25]. Since NdFeB is very sensitive to oxygen and water vapor, the magnetic properties precisely reflect degradation effects pertaining to the particles or the Al_2_O_3_ ALD layer during post-processing. Of particular interest are the remanence B_r_ and the intrinsic coercivity H_ci_. Since B_r_ depends on the amount of magnetic material, this value represents a measure of the mold-filling quality. H_ci_ is an intrinsic material property, and does not depend on the volume of a magnet. Deviating values of B_r_ and H_ci_ indicate an incomplete solidification of the particles or a degradation of the magnetic material due to manufacturing processes [20].

However, VSM measurements are comparatively slow, and necessitate dicing of the substrate into chips. For in-line process monitoring, non-destructive test methods are needed. Qualitative magnetic field measurements with sufficient lateral resolution at the wafer level can be performed by means of magneto-optical microscopy. With this measurement technique, an optical contrast based on the local magnetization is generated due to the Faraday effect. In this way, a non-destructive, fast, and automated optical and magnetic quality control within the process chain can be realized. Figure 4a,b display optical and magneto-optical images, respectively, of integrated NdFeB PowderMEMS microstructures. The images were acquired after wafer-level magnetization at 3.5 T. The resolution in magneto-optical mode is sufficient to resolve magnetic structures with lateral dimensions of 125 µm, as shown in Figure 4c. The qualitative magnetic contrast and optical inspection, including 3D topologies, enable automated and rapid quality control within the process chain, along with identification of defective magnetic microstructures.

Depending on the application requirements, and to ensure process stability, quantitative characterization of magnetic microstructures at the substrate level can be necessary. For this case, a Hall sensor setup was developed that can scan across the wafer surface at close distance. In this way, magnetic fields of test structures can be quantitatively characterized with sufficient lateral resolution to map features with dimensions of 500 µm, as shown in Figure 5.

## 3. Morphology of the PowderMEMS Microstructures

The 3D microstructures fabricated using the PowderMEMS technology represent a novel composite material consisting of three phases: a framework of particles that are in mutual contact, an open network of interconnected voids (pores) filled with a gas or fluid, and a continuous ALD layer separating the first two phases from one another.

Figure 6 displays SEM micrographs of an empty micro-mold pattern formed in a silicon substrate, and an identical pattern of molds with 3D microstructures agglomerated from NdFeB powder (D50 = 5 µm) by 75 nm of ALD-Al_2_O_3_ at 75 °C.

To visualize their morphology, a sample similar to the one shown in Figure 6b was exposed to XeF_2_ vapor to remove the surrounding substrate material. Since XeF_2_ etches silicon isotopically, the substrate material is simultaneously removed on both the top and the bottom, as illustrated in Figure 7a, exposing the 3D microstructures. SEM micrographs of the exposed structures are displayed in Figure 7b,c. From the bottom, only the ALD-Al_2_O_3_ layer is visible, since it covers not only the particles, but also the inner surfaces of the Si molds. At higher SEM acceleration voltages, the Al_2_O_3_ shell is penetrated by the electron beam, and the framework of particles beneath becomes visible (see Figure 7d).

The continuous ALD layer encloses the individual particles entirely throughout the whole volume of the particle bed, keeping them in place and providing considerable mechanical stability to the porous 3D microstructure. Figure 8 illustrates this in more detail, displaying SEM micrographs of a focused ion beam (FIB) cross-section through a NdFeB-based 3D microstructure. In Figure 8b, the ALD-Al_2_O_3_ layer around the particles is especially apparent.

To verify that neighboring particles within the framework were in mechanical point contact with one another, 3D microstructures were agglomerated from Si powder. The top surface of the substrate was then dry-polished to remove a few micrometers of both the substrate material and the solidified structure, as indicated by the blue dashed line of the left-hand cross-section in Figure 9a. During subsequent etching in XeF_2_ vapor, both the silicon of the substrate and the particles, exposed by the polishing, were attacked. The right-hand cross-section in Figure 9a illustrates this principle, while the SEM micrographs in Figure 9b,c show the results. Within a zone of 5 µm from the surface, the Si particles were removed, leaving the continuous ALD layer as an empty, free-standing Al_2_O_3_ shell. The contact points between neighboring particles remained free of Al_2_O_3_, as observed in [19]. Through those “bottlenecks”, the XeF_2_ proceeded from particle to particle during etching. From a depth of approximately 5 µm below the surface, Si particles were still present that had not been etched during the limited exposure time to XeF_2_ vapor.

Figure 10 presents SEM micrographs of structures agglomerated from different powders using 75 nm ALD-Al_2_O_3_; the particles are arranged randomly. Since no mechanical pressure is applied during agglomeration, they remain unchanged in shape and size, and the pores in between the particles are preserved. As the micrographs indicate, all pores are interconnected, forming an open network of voids throughout the structure. The mean pore size and the fill factor mainly depend on the shape and size of the particles. The impact of the nanometer-thin ALD layer on the pore volume is negligible considering the micron-sized particles.

The specific morphological features of the PowderMEMS microstructures without doubt strongly impact their properties. Material parameters such as thermal or electrical conductance, mechanical stability, or the coefficient of thermal expansion cannot be simply derived from the corresponding properties of the bulk material and the ALD. Since suitable simulation models are not yet established, the material parameters must be obtained experimentally. In past research, the main attention has been paid to the durability of PowderMEMS microstructures and their integrability into MEMS processes. The evaluation of application-related properties is subject to both ongoing and future work.

## 4. Applications

The wide choice of materials and customizable properties opens a multitude of applications for the PowderMEMS technology. Next to the shape and volume of the 3D structure itself, additional application-specific physical and chemical characteristics can be controlled by choice of the particles and the ALD layer (see Figure 11). Table 1 presents a list of the applications discussed in this section. The list is not conclusive, and provides application examples in which PowderMEMS enables the improvement and the realization of existing and novel MEMS devices, respectively.

### 4.1. Magnetic Applications

By choosing magnetic particles, PowderMEMS can be used to create integrated microstructures with magnetic properties without the harsh temperatures required for sintering, while also providing the magnetic particles with a passivating ALD layer. These precisely defined structures can therefore be directly integrated into MEMS at the wafer level, replacing cumbersome and expensive hybrid integration of magnetic materials at the chip level. More information on the general usage of micromagnets in MEMS applications can be found in [27].

#### 4.1.1. Integrated Permanent Magnets

High-power permanent micromagnets are of interest for MEMS actuators as well as for sensors. However, until now, a suitable fabrication method at the wafer level that is BEOL-compatible was not available. Figure 12 presents photographs of integrated NdFeB micromagnets created using PowderMEMS. Figure 12a shows an 8-inch Si substrate with micromagnets of various designs. Figure 12b shows triangular silicon frames with micromagnets in the corners, sticking magnetically together and to a metal plate. Magnetization of the micromagnets was performed out of plane at the wafer level with a magnetic field of 3.5 T on a custom 8” wafer magnetization tool.

#### 4.1.2. Energy Harvesting

Figure 13 depicts piezoelectric vibrational energy harvesters with an integrated micromagnet array as a movable tip mass. With such a device, electrical energy can be generated by mechanical excitations—e.g., vibrations and shocks—as well as from time-varying external magnetic fields using magnetic force coupling [23,28]. Two major advantages can be identified compared to devices with conventional silicon-based tip mass: Firstly, increased seismic masses lead to lower resonance frequencies, which better match to the sources of mechanical vibrations available in the environment. Secondly, with a magnet as the seismic mass, energy can be harvested contactlessly from rotational and translational motions. The available magnetic forces are often equivalent to an acceleration of several tens of g’s, if compared to mechanical excitation. Thus, frequency upconversion from both strong mechanical as well as magnetic-pulse-like excitations can be utilized, making it possible to harvest energy at low excitation frequencies of <50 Hz with power outputs of >70 µW [26]. Notably, it was further demonstrated that the generated power is sufficient to switch an NMOS and to initiate near-zero-power wake-up [29]. Apart from that, magnetic field detection with an ultralow limit of detection of 7.2 pT/Hz^0.5^–8.5 pT/Hz^0.5^ in resonance has been demonstrated [30].

#### 4.1.3. Inductors

MEMS inductors with a soft magnetic 3D core are without doubt indispensable for future power electronics. Cores based on PowderMEMS 3D microstructures, as exemplarily shown in Figure 14, were described for the first time in [31]. The impact of different soft magnetic powders was evaluated, and a boost converter with a GaN FET was developed to prove the functionality of the PowderMEMS inductors using a demonstrator with manually wound wire. At 20 MHz, an input voltage of 15 V could be boosted to 25 V on the output at a load current of 481 mA with an efficiency of 87%. It is expected that cores based on porous 3D microstructures will exhibit an outstandingly high thermal stability since, in contrast to other powder-based techniques, no organic materials are involved in the fabrication process and, consequently, in the core.

### 4.2. Optical Applications

Since the very thin ALD layers used for PowderMEMS are typically optically transparent, incident light can easily penetrate deep into a 3D microstructure made from transparent particles. By using optically active particles, this feature can be exploited to develop innovative optical MEMSs.

A first evaluation of pixelated MEMS luminescence converter arrays for laser-based lighting applications, agglomerated from phosphor particles, has been previously described (Figure 15) [32]. For a general description of this lighting principle, the reader is referred to [33]. The quantum efficiency was found to be ~95% in the investigated blue spectral range, which is a value comparable to commercial phosphor converters. A unique feature is the high optical contrast between illuminated and non-illuminated adjacent pixels, differing by around two orders of magnitude. Since all pixels are separated from one another by the substrate material, the light scattering from illuminated pixels into the neighboring ones (optical crosstalk) is strongly suppressed.

### 4.3. Thermal Applications

The tunable intrinsic porosity of PowderMEMS structures can be exploited for both cooling and thermal insulation of MEMSs. Cooling can be typically achieved by flowing a coolant through a 3D microstructure made from particles with high thermal conductivity. On the other hand, if particles with low thermal conductivity are chosen, thermally insulating structures can be realized in MEMSs.

#### 4.3.1. Cooling of MEMSs

Porous 3D microstructures, whose particles generate heat during operation, can be actively cooled by a fluid flowing through the structure. In Figure 16a, this concept is illustrated for the case of a luminescence converter. As already demonstrated, even a slow air flow causes a considerable decrease in the phosphor temperature [32]. With optimized converter geometry and a suitable cooling fluid, the light density of luminescence converters, commonly limited by thermal quenching of the phosphor, could be increased significantly.

Apart from luminescence converters, porous 3D microstructures agglomerated from particles with high thermal conductivity—such as Si or diamond—could be of interest for active cooling of electronic devices (see Figure 16b). In [34] it is shown that the cooling of a heat source by a fluid, flowing through a channel close to the heat source, is improved by a porous medium within this channel. In [35], the thermal conductivity of the underfill between stacked chips is improved by loading the gap with particles with high thermal conductivity prior to the application of the underfill material. Similar features can be achieved with PowderMEMS, in a manner well compatible with most other MEMS processes.

#### 4.3.2. Thermal Insulation

The cumulative thermal conductivity of a porous structure is the sum of the thermal conductivities of the solid structure and the fluid contained within the pores. In a dry system of solid particles and gas, the gaseous phase controls the overall effective thermal conductivity [36]. The heat transfer from particle to particle is strongly hindered, since the area of the point-type contacts between neighboring particles is very small compared to the particle surface, which is in contact with the gas. The effective thermal conductivity can be further decreased by one order of magnitude if the pressure of the gas within the pores is reduced below 1 mbar [36]. Similar pressure values are common working points for plasma-enhanced chemical vapor deposition (PECVD) processes. Accordingly, with optimized geometry and optimal particles, porous 3D microstructures with outstandingly low effective thermal conductivity can be generated using standard MEMS fabrication processes.

Figure 17 illustrates two possible concepts schematically. A finished conventional device, comprising a thin membrane with a heater, electrically connected to bond pads, can be equipped with a porous 3D microstructure from its backside. Subsequent deposition of a sealing layer on the backside—for example, by PECVD—ensures reduced pressure (vacuum) inside it (Figure 17a). Alternatively, the porous 3D microstructure can be realized first; after its planarization and sealing under vacuum, the MEMS device is post-processed on top of the substrate (Figure 17b). This second approach completely replaces the need for a membrane, and would provide a thermally insulating, mechanically robust support that is expected to have a lower failure rate during substrate processing than stress-compensated thin-film membranes, as well as much higher resistivity to overpressure and shocks of the fabricated MEMS device during operation. Moreover, post-processing of through-silicon vias could be used to separate the electrical contacts (bond pads) from the sensing environment by relocating them to the backside of the substrate.

### 4.4. Porous Solid Phases for Microfluidic Applications

A multitude of microfluidic applications can take advantage of the large inner surface area and irregular channel network of PowderMEMS microstructures. The two most basic use cases are as integrated filters for particle retention in fluidic streams, and to enhance the mixing of two or more fluids. Mixing has been shown to greatly benefit from porous microstructures for turbulence induction in low-Reynolds-number flow, thereby shortening the mixing length and reducing mixing time [37,38].

The very large inner surface area of PowderMEMS structures can be used as a solid phase for the adsorption or immobilization of molecules. Typical examples include their use in microfluidic devices for nucleic acid purification via solid-phase extraction, or in devices for miniaturized chromatography [39]. Figure 18 shows a photo of a Si substrate with 20 mm long, meandrous porous structures for on-chip liquid chromatography agglomerated from oxidized SiO_2_ beads with ALD-Al_2_O_3_. Following the structural agglomeration, SiO_2_ was deposited by ALD in the same way to achieve an inner surface suitable for hydrophilic interaction liquid chromatography (HILIC). On-chip separation of a mixture of acetaminophen and gentisic acid was successfully demonstrated [23]. Furthermore, the BEOL compatibility of the PowderMEMS process makes it possible to integrate electrochemical or optical detectors for the direct measurement of the eluted target substances with the microscale chromatography column, forming a complete analysis chip.

Porous solid phases also have important applications in chemical synthesis as a support for immobilized molecules and catalytic surfaces. An example for the immobilization of molecules is the synthesis of nucleic acids in microfluidic devices, where the growing DNA or RNA strand is immobilized on a porous solid support [40].

Very large catalytic surfaces can be achieved by coating the inner surface of a porous 3D microstructure with catalytically active ALD layers such as TiO_2_ or Pt. Here, a key advantage of PowderMEMS is that further properties of the support material can be chosen freely. For example, by selecting an optically transparent powder that is embedded in an optically transparent microchannel, photocatalytic microreactors can be developed that offer even larger catalytic surfaces than those that are achievable by deposition of the catalytic surface on the channel walls alone [41]. To achieve even more complex (organic) catalysts, the surface may then be further modified by the use of molecular layer deposition (MLD) techniques [42]. As MLD can be performed using the same equipment as ALD, PowderMEMS has the unique advantage that both the formation of the porous support framework and the deposition of highly complex catalytic surfaces can be performed as sequential processes in the same reactor.

### 4.5. Sensors

#### 4.5.1. Flow Sensors

Calorimetric flow sensors and pyroelectric sensors commonly use thin-film membranes to achieve thermal insulation of the sensing elements from the bulk substrate [43,44]. For a general description of this type of sensor, the reader is referred to [45]. These membranes are very fragile, which leads to difficulties during sensor fabrication, as well as a high risk of catastrophic failure due to overpressure or physical shock. As already described in Section 4.3.2, PowderMEMS can be used to improve upon existing sensor designs via the formation of a porous 3D microstructure with very low thermal conductivity within the cavity below the membrane (see Figure 17a). Apart from a better mechanical stability in such a way, parasitic effects such as air convection within the backside volume—causing a distortion of the sensor signal—can be suppressed. Enhanced sensing performance is expected in fluids with high thermal conductivity.

A first proof of the thermal insulation properties of porous 3D microstructures, fabricated from particles of different materials using flow sensor designs, is exemplarily shown in Figure 19. Please note that in contrast to Figure 17a, the porous 3D microstructure remained unsealed from the backside in order to allow a variation in the gas pressure within the pores.

#### 4.5.2. Gas Sensors

Since wafers with embedded porous 3D microstructures can be post-processed using common MEMS and semiconductor processes, there are different ways to integrate heaters and electrodes on top, nearby, or even within them. A porous 3D microstructure modified by ALD with a semiconductive gas-sensitive layer such as TiO_2_ can thus be both contacted electrically as well as heated to form a catalytic gas sensor (Figure 20a). For a general description of this type of sensor, the reader is referred to [46]. As is known, the activation energy for the gas-sensitive layer can also be delivered via optical excitation. PowderMEMS offers the unique advantage of using optically transparent particles to obtain the 3D microstructure supporting the catalytic layer [47]. In this way, deep penetration of light into the gas-sensitive structure can be achieved, which is expected to improve the sensitivity (Figure 20b).

Other types of gas sensors could also benefit from the PowderMEMS process. For example, existing paramagnetic oxygen sensors, which rely on macroscale hybrid integration of rare-earth magnets, could be modified to include micromagnets that are integrated at the wafer level, making the post-processing of single chips and hybrid mounting of conventional magnets unnecessary [48].

Another promising application that is relevant for gas sensors is their protection from dust or moisture. Several groups have shown this principle at the chip- or package level, demonstrating that very thin structures that are still mechanically robust are needed in order to prevent unacceptable delays in sensor response [49,50]. With PowderMEMS, thin, microporous, protective caps can be fabricated at the wafer level and integrated with the MEMS devices using common wafer-level bonding techniques (Figure 21). If necessary, the cap wafer can be provided with moisture protection by depositing a corresponding material on its top side before or after bonding.

#### 4.5.3. Electrochemistry and Biosensors

Electrodes for electrochemical applications benefit from large surface areas. For a general description of this type of sensor and its applications, the reader is referred to [51]. With PowderMEMS, there are two possible ways to manufacture high-surface-area electrodes: The first way is ALD of a catalytic material such as Pt onto the inner surface of the porous structure (Figure 22a); however, ALD processes for the deposition of precious metals often use expensive precursors, and are difficult to control [52]. The second strategy is the formation of a powder-based electrode via physical retention of a conductive, catalytic powder inside an etched microcavity on top of a thin-film electrode [53,54]. The retention can be achieved via partial agglomeration of the powder inside the cavity, essentially forming a porous plug on top of the loose powder (Figure 22b), as has already been demonstrated in [22]. A further advantage of this type of electrode is its self-wicking behavior, which enables passive filling of the electrode cavity.

A porous microstructure can also be used as a solid support for the immobilization of biomolecules such as enzymes or antibodies. Hereby, the sensitivity of biosensors can be improved by providing a much larger surface for interaction between the sample containing the target molecules and the immobilized, captured molecules [55,56,57].

Another concept is the utilization of thin-film electrodes, deposited onto a 3D microstructure, for electrochemical sensing. The irregular, rough surface of the 3D microstructure not only causes an effective increase in the electrode area—open “undercut” pores remain within the metal layer due to the limited step coverage of deposition processes such as sputtering or evaporation, so that a fluid can be passed through both the porous 3D microstructure as well as the thin-film electrode on top. In this way, in contrast to common planar arrangements, the entire fluid volume contacts the electrode. Figure 23 illustrates this novel type of flow-through 3D sensor schematically, and presents an SEM micrograph of such a structure from the top side.

## 5. Conclusions and Outlook

The current state of the novel PowderMEMS microfabrication technology was reviewed alongside a discussion of potential applications and necessary characterization techniques. The possibility of integrating BEOL-compatible 3D microstructures at the substrate level from a wide choice of materials yields innovation potential for microsystems in numerous fields of application. It has been shown how the magnetic, thermal, optical, mechanical, and fluidic properties of integrated 3D microstructures can be tailored to cater for functions such as sensing, actuation, inductance, insulation, filtering, or cooling.

Implementation of the PowderMEMS technology on an industrially relevant scale was demonstrated using standard 8-inch Si substrates. The already-available fabrication line supplements an established BEOL cleanroom, and includes the dry filling of the powders, batch ALD solidification, and substrate conditioning, as well as wafer-level characterization and magnetization.

Further research is needed in order to determine the effective properties of the diverse PowderMEMS microstructures. Since suitable simulation models are not yet available, they must be determined experimentally. So far, the main attention has been paid to the durability of PowderMEMS microstructures embedded in silicon substrates, because of their high relevance to the MEMS/semiconductor industry. For example, the resistance to thermal treatments and O_2_ plasma has been investigated, and thermal cycling tests have been executed to assess mechanical degradation effects. In addition, the thermal conductivity of selected microstructures has been evaluated to a limited extent. Other properties—such as hardness or modulus of elasticity, the thermal coefficient of expansion, or electrical conductivity—have not yet been determined.

The authors invite the MEMS community to embrace the PowderMEMS technology, and are optimistic that further research into different powder and ALD layer combinations will lead to a multitude of novel applications.

## 6. Patents

Granted patents and application publications related to the PowderMEMS technology are summarized in Table 2 below.

## Figures and Tables

**Figure 1 micromachines-13-00398-f001:**
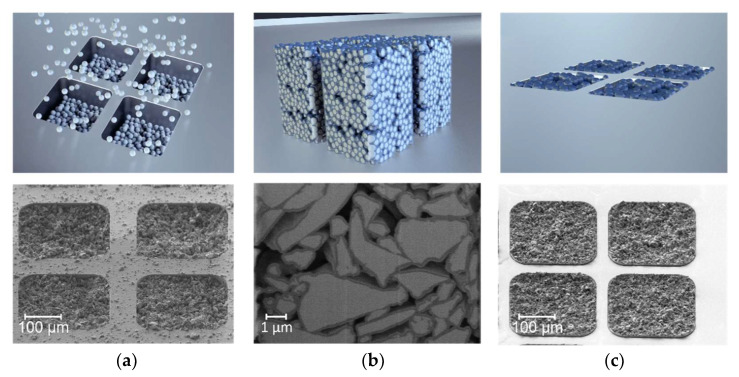
Illustration of the PowderMEMS process [19] by schematic drawings (**top**) and corresponding SEM micrographs of real samples, obtained from NdFeB powder (**bottom**): (**a**) Dry-filled micro-molds with particles. (**b**) Solidification of the powder bed via ALD. The blue layer in the drawing represents the ALD layer (not to scale). The surrounding substrate is not shown for clarity. In the SEM micrograph the particles (NdFeB) appear bright grey, while the surrounding ALD layer (75 nm Al_2_O_3_) is dark grey. (**c**) Removal of excess particles from the substrate surface to enable post-processing in a cleanroom environment.

**Figure 2 micromachines-13-00398-f002:**
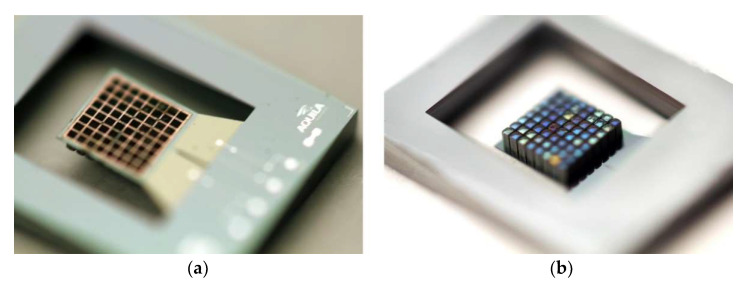
(**a**) Top side of a piezoelectric energy harvester with permanent wafer-level integrated NdFeB magnets (black pixels). (**b**) Bottom of the silicon cantilever exhibiting the magnetic PowderMEMS structures. Each magnetic pixel is 180 × 180 × 500 µm^3^ in size. Refer to Section 4 and [26] for details.

**Figure 3 micromachines-13-00398-f003:**
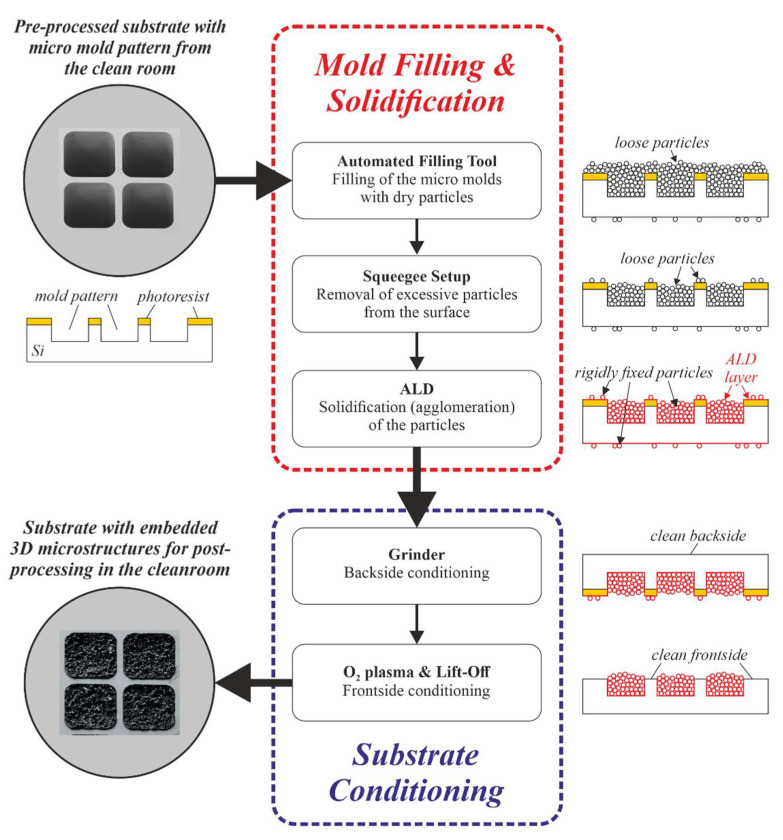
Schematic illustration of the realization of embedded functional 3D microstructures on 8-inch Si substrates, as implemented at Fraunhofer ISIT. All process steps are carried out using automated equipment, enabling industrially relevant manufacturing.

**Figure 4 micromachines-13-00398-f004:**
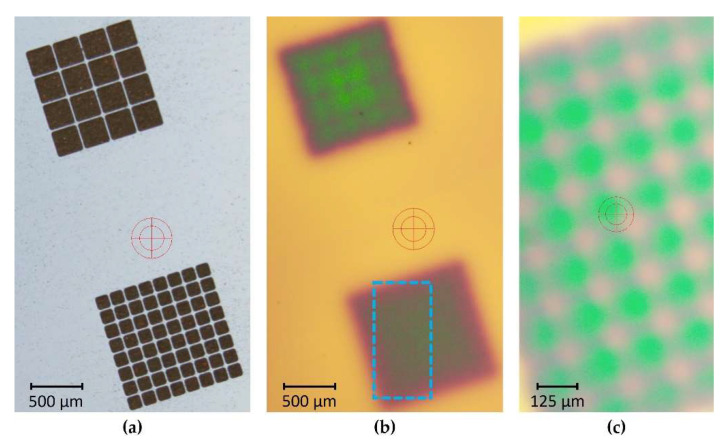
(**a**) Optical image of NdFeB microstructures embedded in a silicon substrate. (**b**) The same structures imaged with a magneto-optical sensor based on the Faraday effect. The contrast corresponds to the strength of the out-of-plane component of the magnetic field. (**c**) Magneto-optical image of the ROI marked by a dashed frame in (**b**), acquired at higher magnification with a different lens.

**Figure 5 micromachines-13-00398-f005:**
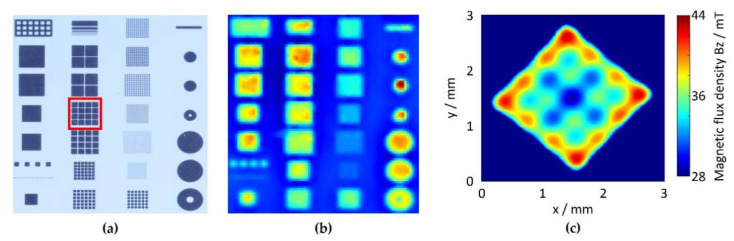
(**a**) Microscope image of NdFeB microstructures embedded in a silicon substrate. (**b**) Scan of the same area with a 3D Hall sensor. The color code represents magnetic flux perpendicular to the substrate surface. (**c**) Detailed scan of the 4 × 4 array marked in red in (**a**), consisting of (500 × 500) µm^2^ magnets. The vertical distance between the Hall sensor and the substrate surface was 120 µm.

**Figure 6 micromachines-13-00398-f006:**
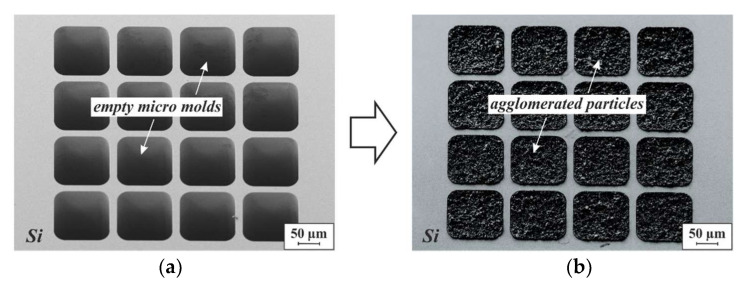
(**a**) SEM micrograph of a micro-mold pattern etched into a silicon substrate with DRIE. The mold cavities are ~400 µm deep and 125 µm × 125 µm in size. (**b**) SEM micrograph of 3D microstructures agglomerated from NdFeB powder with 5 µm mean particle size by 75 nm ALD-Al_2_O_3_, using the mold shown in (**a**).

**Figure 7 micromachines-13-00398-f007:**
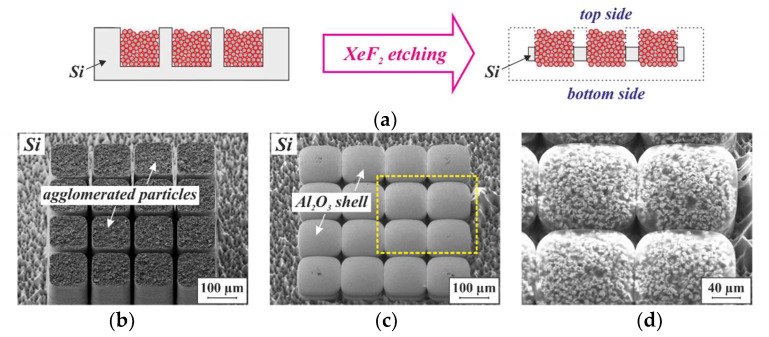
(**a**) Schematic illustration of the isotropic etching in XeF_2_ vapor, applied to expose 3D microstructures that are embedded in a silicon substrate. The black dotted frame on the right-hand cross-section indicates the initial dimensions of the silicon chip. SEM micrographs of the (**b**) top and (**c**) bottom sides of the NdFeB 3D microstructures shown in Figure 6 after release etch by XeF_2_ vapor. Both micrographs were obtained with an acceleration voltage of 2 kV. (**d**) SEM micrograph of the area within the yellow dashed frame in (**c**), obtained with an acceleration voltage of 20 kV.

**Figure 8 micromachines-13-00398-f008:**
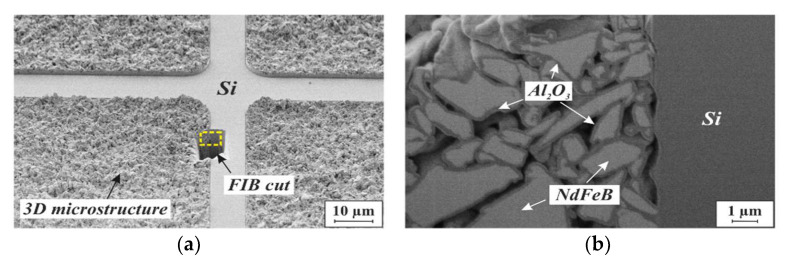
SEM micrographs of an FIB cut through a porous 3D microstructure agglomerated from NdFeB powder with 5 µm mean particle size by 75 nm ALD-Al_2_O_3_: (**a**) FIB preparation overview. (**b**) Detailed side view of the FIB cut area within the yellow frame in (**a**).

**Figure 9 micromachines-13-00398-f009:**
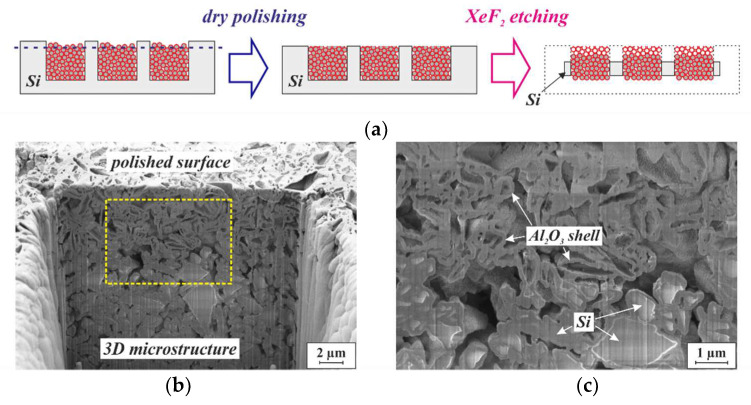
(**a**) Schematic illustration of the procedure applied to investigate the contact points between neighboring particles. (**b**) SEM micrograph of an FIB cut through a 3D microstructure that has been agglomerated from Si particles with 1 µm mean size by 75 nm ALD-Al_2_O_3_. After dry-polishing at the wafer level and dicing into chips, the 3D microstructure was exposed to isotropic etching in XeF_2_ vapor. (**c**) Micrograph of the area within the dashed yellow frame in (**b**).

**Figure 10 micromachines-13-00398-f010:**
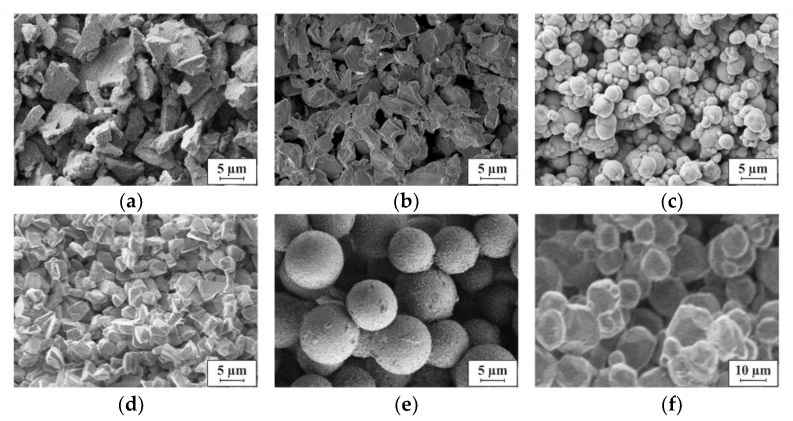
SEM micrographs of porous 3D microstructures solidified by 75 nm ALD-Al_2_O_3_ and made of powders from different materials, with varying mean particle size d: (**a**) d = 9 µm Al_2_O_3_, (**b**) d = 5 µm NdFeB, (**c**) d = 1 µm carbonyl iron, (**d**) d = 5 µm diamond, (**e**) d = 10 µm oxidized silicon beads, (**f**) d = 14 µm phosphor. Between the particles empty voids form a network of pores.

**Figure 11 micromachines-13-00398-f011:**
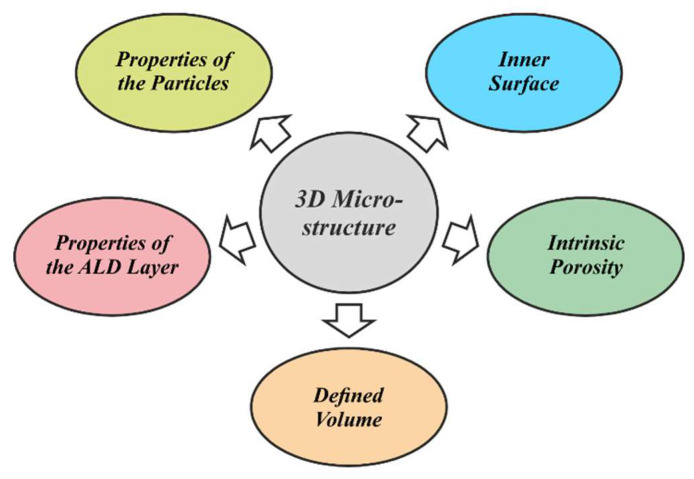
Physical and chemical properties of PowderMEMS 3D microstructures that can be tailored towards specific applications.

**Figure 12 micromachines-13-00398-f012:**
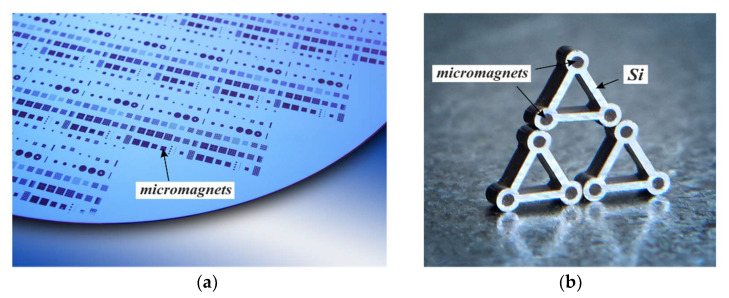
(**a**) PowderMEMS micromagnets fabricated from NdFeB powder inside an 8” wafer. (**b**) Triangular Si frames with magnetized NdFeB micromagnets in the corners, magnetically sticking together and to the underlying metal plate. The Si is ~700 µm thick; the micromagnets are 300 µm in diameter and extend ~600 µm into the Si. A close-up of a NdFeB-based microstructure is provided in Figure 10b.

**Figure 13 micromachines-13-00398-f013:**
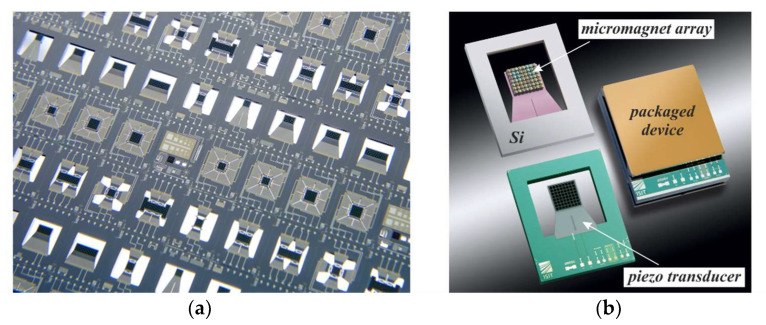
(**a**) Wafer with piezoelectric vibrational harvesters with integrated micromagnet arrays. (**b**) Detailed views of a single energy harvester from the top (lower left) and the bottom (upper left), as well as a packaged device.

**Figure 14 micromachines-13-00398-f014:**
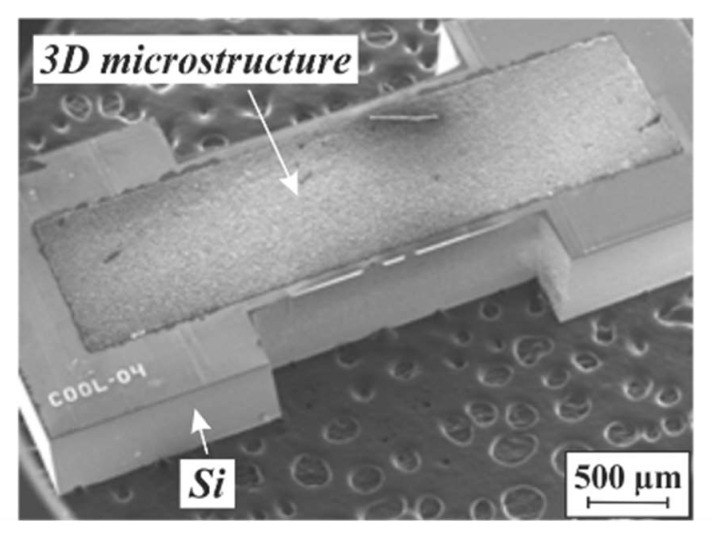
SEM micrograph of a soft magnetic PowderMEMS core for MEMS inductors, agglomerated from Fe particles by 75 nm ALD-Al_2_O_3_ at 75 °C. A close-up of an Fe-based microstructure is provided in Figure 10c.

**Figure 15 micromachines-13-00398-f015:**
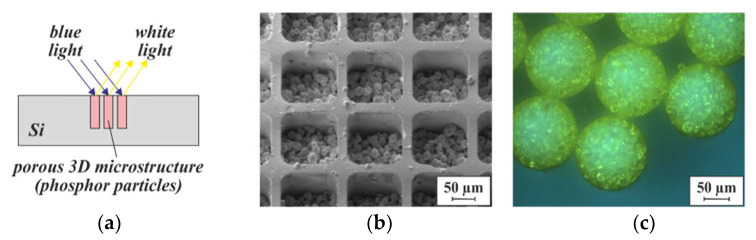
(**a**) Schematic illustration of a PowderMEMS luminescence converter based on agglomerated phosphor particles [32]. (**b**) SEM micrograph of the top side of a converter array, consisting of square-shaped pixels. (**c**) Optical image of the bottom side of another converter array, consisting of circular pixels, after removal of the Si substrate by etching in XeF_2_ vapor. A close-up of a phosphor-based microstructure can be found in Figure 10f.

**Figure 16 micromachines-13-00398-f016:**
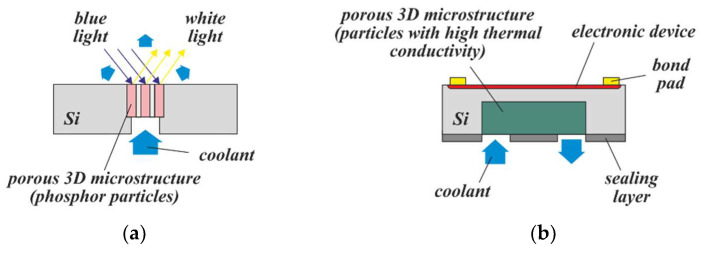
Concepts for actively cooled MEMSs: A coolant is forced through (**a**) a porous luminescence converter, manufactured from phosphor particles, and (**b**) a porous 3D microstructure, manufactured from particles with high thermal conductivity, which is embedded in a Si chip with a heat-generating electronic device.

**Figure 17 micromachines-13-00398-f017:**
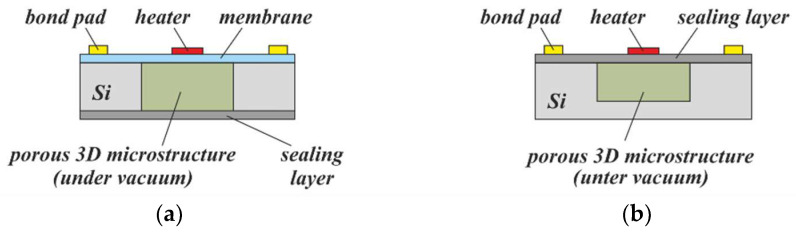
Concepts for the thermal insulation of calorimetric MEMS devices using porous 3D microstructures under vacuum: In (**a**) the 3D microstructure is post-processed within the backside cavity of a finished, membrane-based device and, finally, sealed under vacuum. In (**b**), the 3D microstructure is pre-processed including planarization and sealing under vacuum. Subsequently, the MEMS device is processed on top. Please note that in both cross-sections the electrical wiring from the heater to the bond pads is not shown.

**Figure 18 micromachines-13-00398-f018:**
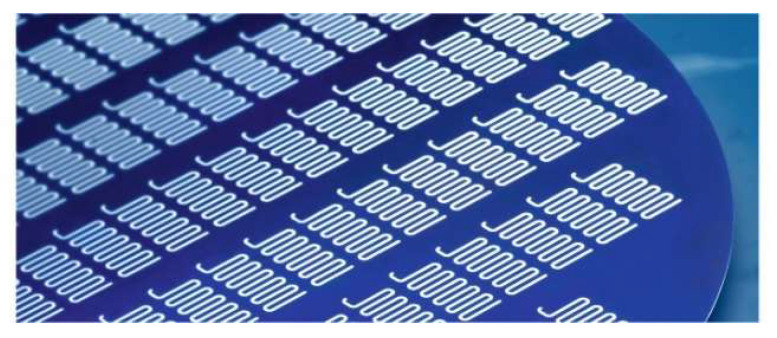
Silicon substrate with 20 mm long, meander-type porous 3D microstructures, agglomerated from monodisperse oxidized Si beads with a mean size of 10 µm using 75 nm ALD-Al_2_O_3_. A close-up of a 3D microstructure agglomerated from Si beads is provided in Figure 10e.

**Figure 19 micromachines-13-00398-f019:**
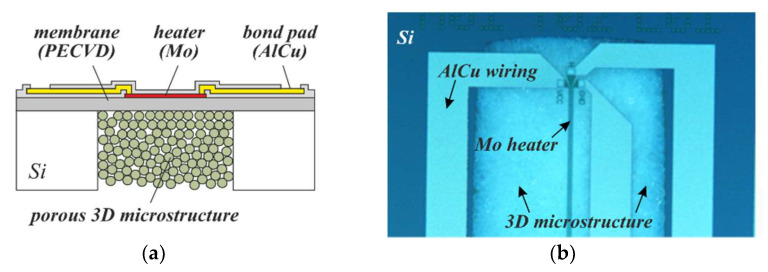
(**a**) Cross-section of a thin-film-membrane-based calorimetric flow sensor, whose backside cavity has been equipped with a 3D microstructure at the wafer level. (**b**) Optical image of the top side of a particular sensor design. The 3D microstructure can be seen through the transparent membrane.

**Figure 20 micromachines-13-00398-f020:**
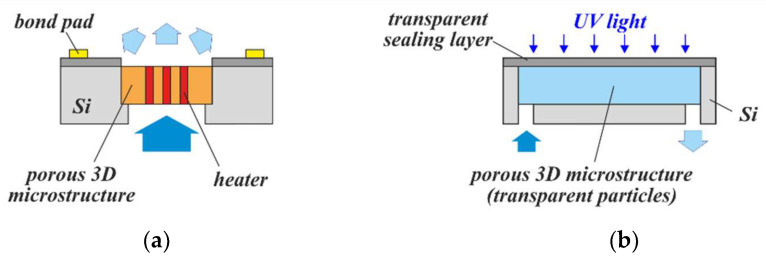
Concepts for PowderMEMS gas sensors: (**a**) Resistive heating of the catalyst. (**b**) Optical activation of the catalyst.

**Figure 21 micromachines-13-00398-f021:**
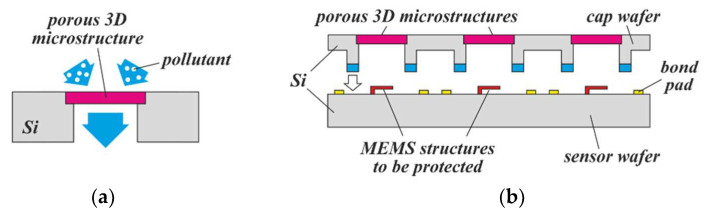
(**a**) Concept of an environmental protection cap for gas sensors, which must communicate with the environment, but degrades if contaminated with particles or moisture; (**b**) illustration of their integration with sensors by wafer-level packaging.

**Figure 22 micromachines-13-00398-f022:**
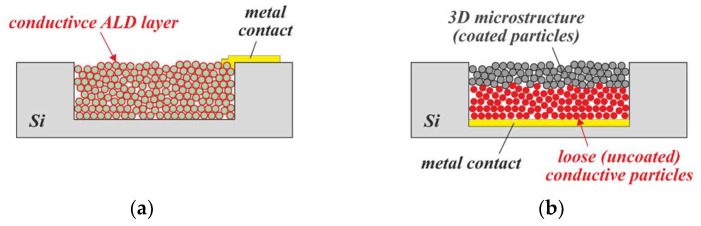
Concepts for large-surface-area electrodes manufactured via the PowderMEMS process: (**a**) Electrode formed by coating of the porous structure with a conductive ALD layer. (**b**) Electrode formed by confining a loose conductive powder inside a cavity.

**Figure 23 micromachines-13-00398-f023:**
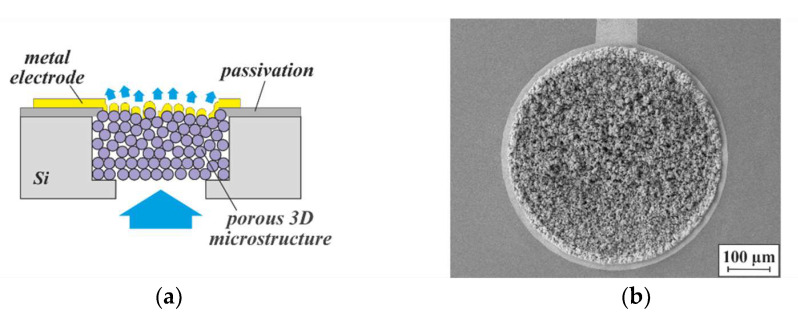
(**a**) Schematic concept of a flow-through 3D sensor with a thin-film electrode deposited on top of a porous 3D microstructure, and (**b**) SEM micrograph of the top side of such a structure. The electrode consists of sputtered Au. The porous 3D microstructure, 500 µm in diameter and ~500 µm deep, was agglomerated from 5 µm NdFeB particles by 75 nm ALD-Al_2_O_3_.

**Table 1 micromachines-13-00398-t001:** Exemplary list of possible applications of the PowderMEMS technology. The individual applications included in this table are presented and discussed in the following text.

	Application	Powder	Powder Property of Interest	ALD Layer	ALD Layer Property of Interest
Magnetic	Integrated permanent magnets	NdFeB, Fe, ferrites		Al_2_O_3_	
	Energy harvesting		Hard/soft ferromagnetism		Mechanical
	Inductors				
Optical	Light conversion	Phosphor	Fluorescence	Al_2_O_3_	Mechanical/optical transparency
Thermal	Cooling of MEMS	Si	High thermal conductivity	Al_2_O_3_	Mechanical
	Thermal insulation	Pyrogenic SiO_2_	Low thermal conductivity	Al_2_O_3_/SiO_2_	Mechanical
Fluidic	Filter				Mechanical
	Mixer	Si/SiO_2_/Si_3_N_4_	Mechanical	Al_2_O_3_/SiO_2_	
	Solid support				Adsorption of (bio)molecules
Sensors	Flow sensors	Pyrogenic SiO_2_	Low thermal conductivity	Al_2_O_3_/SiO_2_	Mechanical
	Gas sensors	Si/Metal	Electrical conductivity	TiO_2_	Catalysis
	Electrochemistry/biosensors	Si/metal/glassy carbon	Electrical conductivity	Metal	Electrical conductivity/adsorption of (bio)molecules

**Table 2 micromachines-13-00398-t002:** List of granted patents and application publications related to PowderMEMS.

No.	Granted Patents and Publications	Short Description of the Patent Family
**1**	EP2670880B1 US9221217B2 JP6141197B2	Fabrication of porous 3D microstructures, basic method
**2**	EP3234968B1 US20170278605A1	Utilization of magnetic 3D microstructures for actuators and inductors
**3**	CN107980010B US2018029002A1	Application of the large inner surface of porous 3D microstructures
**4**	US10647915B2	Miniaturized luminescence converter based on porous 3D microstructures
**5**	EP3284714B1 CN107761069B US2018051308A1	Partial agglomeration, movable parts in a closed cavity utilizing porous 3D microstructures
**6**	DE102016215616B4 US10854223B2	Magnetic scale based on porous 3D microstructures
**7**	US11137364B2	Thermal isolation based on porous 3D microstructures
**8**	WO2020128018A1	Force transfer based on magnetic interaction of porous 3D microstructures
**9**	DE10201901744B4 US2020178000A1	Magnetically boosted MEMS loudspeaker
**10**	DE102019210177B4 US20210082611A1	Micromagnet arrays with opposite magnetization
**11**	WO21028345A3	Magnetic position detection for MEMS based on porous 3D microstructures

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
