# Peer review of "PowderMEMS—A Generic Microfabrication Technology for Integrated Three-Dimensional Functional Microstructures"

_micromachines, 2022, doi:10.3390/mi13030398_

Round 1

Reviewer 1 Report

The manuscript discussed the fabrication technology of the functional microstructures. They also showed the applications of microstructures. The topic is interesting, but there is still something that needs further improvement. First, the authors should list some tables to summarize by using the nanostructure to assist the applications, such as the table of the gas sensor in sensitivity, target gas, or something. Second, the applications of the existing cases are few, the authors can list more cases to show the applications. Third, the authors can show the advantage and disadvantages of Microfabrication Technology. More information on these micromaterials needs to summarize in this manuscript as well. In all, I recommend accepting this manuscript after the authors solve the above-mentioned questions. 

Author Response

Please find our reply in the attached Word document.

Reviewer 2 Report

The authors present a very interesting overview of the novel compatible microfabrication technology of 3D Powder MEMS. They detail some interesting applications in the field of magnetic, optical, thermal, electrochemical and microfluidic MEMS.

In the Figure 13 b, have to replace  "unter" with under.

Author Response

(The authors gave the same response as above.)

Reviewer 3 Report

  1. the authors should carefully organize this paper.  In chapter 2 and chapter 5.1, they relate to the core procedure, which is confusing to the readers. Please focus one chapter only on the fabrication process.
  2. the authors should mention some challenges or unsolved problems about powderMEMS, which is helpful for other researchers. 
  3. There are many applications for powderMEMS. When reading your papers, only see the microstructures. The principles underlying these applications should be mentioned. 
  4. Please make a table about the categories of the powders and their corresponding applications. 

Author Response

(The authors gave the same response as above.)

Round 2

Reviewer 1 Report

The manuscript was have solved all my issues. I recommend publishing in the current form.

Reviewer 3 Report

The authors carefully addressed my questions and thus I suggest accepting.